# HypeR: Multitask Hyper-Prompted Training Enables Large-Scale Retrieval Generalization

**Zefeng Cai**[1][*] **Chongyang Tao**[2], **Tao Shen**[2], **Can Xu**[2], **Xiubo Geng**[2], **Xin Lin**[1], **Liang He**[1]
**Daxin Jiang**[2][†]
Department of Computer Science, East China Normal University[1]
Microsoft Corporation, Beijing, China[2]
`oklen@foxmail.com`, `{lhe,xlin}@cs.ecnu.edu.cn`,
`{chotao,shentao,caxu,xigeng,djiang}@microsoft.com`

## Abstract

Recently, large-scale text retrieval has made impressive progress, facilitating both information retrieval and downstream knowledge-intensive tasks (e.g., open-domain QA and dialogue). With a moderate amount of data, a neural text retriever can outperform traditional methods such as BM25 by a large step. However, while being applied to out-of-domain data[1] , the performance of a neural retriever degrades considerably. Therefore, how to enable a retriever to perform more robustly across different domains or tasks and even show strong zero-shot transfer ability is critical for building scalable IR systems. To this end, we propose HypeR, a hyper-prompted training mechanism to enable uniform retrieval across tasks of different domains. Specifically, our approach jointly trains the query encoder with a shared prompt-based parameter pool and a prompt synthesizer that dynamically composes hyper-prompt for encoding each query from different tasks or domains. Besides, to avoid the mode collapse of prompt attention distribution for different queries, we design a contrastive prompt regularization that promotes the mode of prompt attention to be aligned and uniform. Through multi-task hyper-prompted training, our retriever can master the ability to dynamically represent different types of queries and transfer knowledge across different domains and tasks. Extensive experiments show our model attains better retrieval performance across different tasks and better zero-shot transfer ability compared with various previous methods.

## 1 Introduction

Large-scale retrieval aims to retrieve *relevant documents* from millions to billions of documents according to a given query, which is the so-called first stage retrieval (Cai et al., 2021). It can benefit for resolving various knowledge-intensive tasks significantly (Guu et al., 2020; Lewis et al., 2020), since the retrieved relevant documents contain explicit knowledge of world (Petroni et al., 2021). Traditional term-matching methods including tf-idf and BM25 (Yang et al., 2017) can effectively achieve retrieval by building an inverted index and perform fairly well regardless of domains, however, recent popular neural retrievers outperform them by a large step with a moderate amount of task-specific data (Karpukhin et al., 2020; Formal et al., 2021b; Khattab & Zaharia, 2020).

For neural retrieval, a common way is to use pre-trained language models (e.g., BERT) (Devlin et al., 2019) to encode queries and documents into vectors respectively, which is known as Bi-Encoder. Although neural retrievers can be optimized effectively by utilizing the samples of specific tasks, in real-world applications, the formats of queries are different and the expected priorities of query vectors are varying considerably from task to task. For example, in Naturals Questions dataset (Kwiatkowski et al., 2019), a query such as "*what was the first capital city of Australia*" is a simple question sentence, however, in Wizard of Wikipedia dataset (Dinan et al., 2018), a query such as "*...Snoop Dogg is so*

---

[*]Work done during the internship at Microsoft.
[†]Corresponding author.
[1]Noting that here out-of-domain data refers to different tasks with different types of queries or the same task with data from different domains.

*awesome, he's a great rapper and does a lot for his community as well...*" contain multiple declarative sentences with implicit retrieval target. Besides the difference in query formats, different tasks also require generating query vectors with different richness or intents, in HotpotQA dataset (Yang et al., 2018) an input query "*which game was published first, Near and Far or King of Tokyo?*" expects an input query that can retriever documents relevant to the two mentioned items which are fair different from the queries in Natural Question that require retrieving specific facts to only one item. Those differences between tasks cause significant performance degradation when a model is applied to different tasks. Moreover, there is also a data sparse problem for recently popular tasks (Almeida & Matos, 2020), which expects a better generalization of a neural retriever (Thakur et al., 2021).

To resolve the above challenges, we aim to build a universal model that is capable to process queries uniformly regardless of the differences between different tasks including varying formats of input queries and the unique features of query vectors for specific tasks. Meanwhile, we expect our model can obtain stronger generalization abilities which can be reflected by promising zero-shot and few-shot performance in large-scale retrieval. Specifically, the first problem is how to enable a universal query process. For a neural retriever, the ability to resolve a specific task means a set of parameters trained on this task. Although one can train different models for each tasks (Karpukhin et al., 2020) or simply use a shared encoder with multi-tasking setting (Maillard et al., 2021), the first method leads to heavy parameter cost while the second method results in potentially indifferent generalization abilities.

To this end, we propose HYPER, a multi-task HYPEr-prompted training mechanism that can be combined with any transformer-based neural Retrieves. HYPER consists of two key components. The first component is **Q**uery-conditional **P**rompt **S**ynthesizer (QPS) that leverages the attention module to synthesize suitable parameters of query encoder for different queries, which enables our query encoder to master the ability to dynamically represent different types of queries and transfer learned parameters across different tasks and domains by multi-task training. Nevertheless, we find merely applying QPS results in a mode collapse problem of attention scores distributions, which causes our query encoder fails to learn different abilities to process queries for different tasks. To deal with this problem, we propose the Contrastive Prompt Regularization (CPR) to encourage the parameter synthesizing of the same tasks to become similar for better training effectiveness while promoting our query encoder to distinguish queries of different tasks and thus avoid mode collapse problems. Through the above multi-task hyper-prompted training, our HYPER can master the ability to dynamically represent different types of queries and transfer knowledge across different domains and tasks. Therefore, HYPER can enable large-scale retrieval generalization in the zero-shot and few-shot scenarios.

To conclude, our contributions are three-fold as follows, i) we present HYPER, a multitask hyper-prompted training mechanism that enables a neural retriever to dynamically process different types of queries with different hyper-prompts and transfer learned knowledge across different domains and tasks. ii) to impede the uniform retrieval in model construction and optimization, we propose **Q**uery-conditional **P**rompt **S**ynthesizer (QPS) along with **C**ontrastive **P**rompt **R**egularization (CPR) to synthesize suitable prompts for different queries. iii) Experiments in zero-shot in-domain and cross-domain retrieval tasks reflect the superior generalization provided by HYPER and the strong multi-tasking performance indicates the achieving of uniform retrieval.

## 2 METHOD

**Task Formation** For the large-scale text retrieval, we aim to seek document $d_+$ containing relevant knowledge from a large collection of documents $D$ to answer the query $q$. Although input queries vary from task to task, we propose employing only one retriever to process them uniformly. Specifically, for datasets $\mathcal{C} = \{\mathcal{T}_1, \mathcal{T}_2, \ldots, \mathcal{T}_t\}$ and out-of-domain data $\widetilde{\mathcal{C}} = \{\mathcal{T}_{t+1}, \mathcal{T}_{t+2}, \ldots, \mathcal{T}_{t+k}\}$,where $t$,$k$ is the number of tasks with training samples and without training samples respectively, the goal is to learn a neural retriever model $P(d_+|q, D; \theta)$ ($\theta$ denotes the parameters of the model) with $\mathcal{C}$ and perform well on these in-domain tasks, while transferring the learned knowledge to process a new query $q$ from out-of-domain datasets $\widetilde{\mathcal{C}}$. Thus, given any queries in $\mathcal{C} \cup \widetilde{\mathcal{C}}$, one can find the proper knowledge documents $d_+$ following $P(d_+|q, D; \theta)$.

**Model Overview** Building upon a pre-trained neural retriever, HYPER aims to dynamically synthesize suitable prefixes to enable the retriever to process different queries uniformly and an illustration

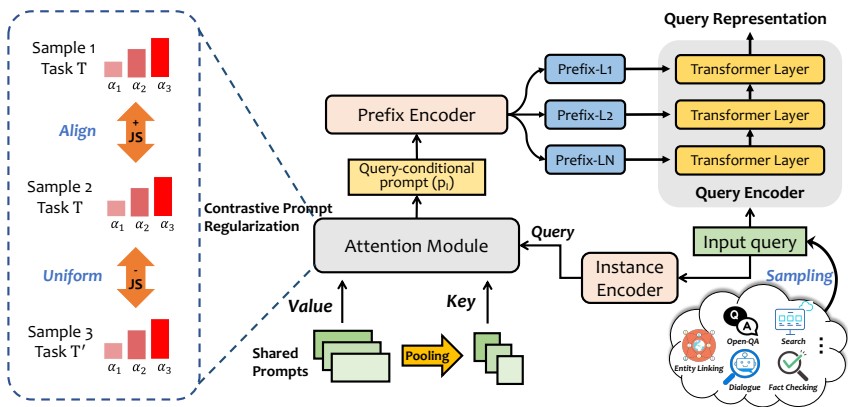

Figure 1: An illustration of HYPER architecture in multi-tasking training.

of it is provided in Figure 1. HYPER first leverage the instance encoder $\theta_I$ to generate the *Query* representation of a query , while hyper-prompts $\mathbf{P}$ are used as *Key* and *Value* for attention module. Therefore, we can dynamically generate different prompts for different types of queries. Besides, our proposed contrastive prompt regularization is used to avoid the mode collapse problem which is crucial for learning different parameters for different types of queries.

## 2.1 PROMPT SYNTHESIZING

There are three main components in our query encoding process including an instance encoder $\theta_I$, shared basic prompts $\mathbf{P} = \{\boldsymbol{p}_i | \boldsymbol{p}_i \in \mathbb{R}^{m \times d}, i \in \{1, \cdots, N\}\}$ where $N$ is the number of shared prompts, $m$ is the length of each basic prompt, and the prefix encoder $\theta_p$. To enable our model to process different tasks uniformly, we store learned knowledge to process different queries into shared prompts and synthesize prompts for different queries dynamically through the attention module. Moreover, we introduce a contrastive prompt regularization to prevent mode collapse of attention scores, which is crucial for HYPER to effectively learn diverse knowledge to synthesize different prompts for queries of different tasks. In the following, we will first describe how HYPER generate the corresponding prefix for different queries. Then, we explain the mode collapse problem in our attention module and how CPR resolves it.

**Query-conditional Prompt Synthesizing**  We intend to generate dynamic hyper-prompts with the global semantic of a query which enables a neural retriever to adapt to different types of queries across different domains. To generate a *query* representation for our prompt attention module, we first map *input query* $q$ into corresponding word embeddings representation $\mathbf{X} = [\boldsymbol{w}_1, \boldsymbol{w}_2, \cdots, \boldsymbol{w}_l] \in \mathbb{R}^{l \times d}$, where $l$ is the length of query, $d$ is the dimension of word embedding. Then we employ max pooling along with the sequence axis of $\mathbf{X}$ and obtain $\hat{\mathbf{X}} = \text{MaxPooling}(\mathbf{X})$. Finally, we utilize a non-linear transformation to generate the incipient query representation as follows:

$$H_I = \textbf{GELU}(\mathbf{W}_1 \hat{\mathbf{X}})\mathbf{W}_2^{\text{T}}, Q = \textbf{LayerNorm}(H_I). \tag{1}$$

Here, $\mathbf{W}_{\{1,2\}} \in \mathbb{R}^{d_h \times d}$ are the transformation matrices and $d_h$ is the dimension of the hidden variable. **GELU** is Gaussian Error Linear Unit (Hendrycks & Gimpel, 2016) and **LayerNorm** is the Layer-wise Normalization (Ba et al., 2016). Similar to query encoding, we employ max pooling operation along with the prompt length axis to transform each prompt into $\hat{\boldsymbol{p}}_i = \text{MaxPooling}(\boldsymbol{p}_i), i \in \{1, \cdots, N\}$. Thus, we can use $Q\hat{\boldsymbol{p}}_i^{\text{T}}$ to imply the fitness of different parameters for an input query. We employ softmax to normalize these scores and form a *prompt attention distribution* $\mathcal{A} \in \mathbb{R}^N$, which is further used for synthesizing the final query-conditional prompt $\boldsymbol{p}_I$. The process is formally described as:

$$\alpha_i = \frac{\exp(Q\hat{\boldsymbol{p}}_i^{\text{T}}/\tau)}{\sum_i^N \exp(Q\hat{\boldsymbol{p}}_i^{\text{T}}/\tau)}, \boldsymbol{p}_I = \sum_i^N \alpha_i \boldsymbol{p}_i. \tag{2}$$

where $\alpha_i$ is the normalized attention score of the $i$-th prompt, $\boldsymbol{p}_I \in \mathbb{R}^{m \times d}$ is generated query-conditional prompt, and $\tau$ is the pre-defined temperature. To further improve the effectiveness of our proposed method, we transform the generated prompt into prefix (Liu et al., 2021a) that owns the better representational ability. Although one can simply employ up projection to generate prefix for each layer of retriever, we find this approach considerably increases the number of total

parameters which may cause the degradation of the generalization. Therefore, we employ a parameter-efficient transformation (Stickland & Murray, 2019; Houlsby et al., 2019) as the prefix encoder $\theta_p = \{\mathbf{W}_{dp}, \mathbf{W}_{up}\}$ to generate the prefix which can be described as follow:

$$\boldsymbol{h} = \mathbf{W}_{dp}\boldsymbol{p}_I, \mathbf{PL} = \mathbf{W}_{up}\mathrm{Tanh}(\boldsymbol{h}) \tag{3}$$

where $\mathbf{W}_{dp} \in \mathbb{R}^{d_p \times d}$ and $\mathbf{W}_{up} \in \mathbb{R}^{Ld \times d_p}$ are projection matrices, $d_p$ is down projection dimension of $\boldsymbol{p}_I$, $L$ is the number of layers of query encoder $\theta_q$, Tanh is the hyperbolic tangent function, $h$ is the intermediary low-dimension representation and $\mathbf{PL} \in \mathbb{R}^{Ld}$ is the dynamically generated parameters that can be split into prefixes for each layer of the neural retriever.

**Contrastive Prompt Regularization** Although the above mechanisms can generate query-conditional prefixes and share parameters across different tasks, we find it results in the so-called mode collapse problem of attention score distributions. Specifically, the attention score distributions of different queries are very similar which may cause our module degenerating to a prefix-tuning. Moreover, we expect queries belonging to the same task generate similar attention scores distribution while queries belonging to different tasks own dissimilar attention scores distribution. To simultaneously learn representations of hyper-prompts and clustering for different types of queries across different domains or tasks, we propose the **C**ontrastive **P**rompt **R**egularization (CPR) that employs soft constraint to cluster the attention scores implicitly and thus avoid the mode collapse problem. CPR can be formally described as follows.

$$\mathcal{L}_{\mathrm{CPR}}(\mathcal{C}) = \sum_{\mathcal{B} \in \mathcal{C}} \sum_{q_i \in \mathcal{B}} \Big( \underbrace{\sum_{q_j \in \mathcal{B}, \mathbb{I}_{q_i} = \mathbb{I}_{q_j}} \mathcal{D}_f\big(\mathcal{A}(q_i), \mathcal{A}(q_j)\big)}_{\text{alignment}} - \underbrace{\sum_{q_k \in \mathcal{B}, \mathbb{I}_{q_i} \neq \mathbb{I}_{q_j}} \mathcal{D}_f\big(\mathcal{A}(q_i), \mathcal{A}(q_k)\big)}_{\text{uniformity}} \Big). \tag{4}$$

Here, $\mathcal{B}$ is a mini-batch of training samples randomly selected from $\mathcal{C}$, $\mathcal{A}(q_*) \sim P(z) = \alpha_z, z \in \{1, 2, \cdots, N\}$ is the attention score distribution of the input query $q_*$, $\mathbb{I}_{q_*}$ is an indicator function that represents the dataset of a task that the query $q_*$ belonging to, $\mathcal{D}_f$ is a divergence that measures the similarity of two distributions. Inspired by contrastive learning (Wang & Isola, 2020), the first term that contains $\mathcal{D}_f\big(\mathcal{A}(q_i), \mathcal{A}(q_j)\big)$ can be viewed as *alignment regularization* that encourages similar queries generated by similar prefixes, and the second term that contains $\mathcal{D}_f\big(\mathcal{A}(q_i), \mathcal{A}(q_k)\big)$ cab be viewed as *uniformity regularization* that prevents mode collapse of distributions of attention score. In our implementation, we use Jensen-Shannon divergence (Manning & Schütze, 2002) since it owns certain upper and lower bounds which avoids the numeric overflow in optimizing.

## 2.2 UNIFORM RETRIEVAL WITH QUERY-CONDITIONAL PROMPT

**Lexicon-weighted Retriever** HYPER is compatible with any deep neural text retriever based on Transformer (Vaswani et al., 2017) architecture, however, we find the lexicon-weighted retrieval method is more promising to attain better zero-shot retrieval[2]. Therefore, we adopt a lexicon-weight language model SPLADE (Formal et al., 2021b) $\mathbf{LM}(\cdot, \cdot; \theta_q)$ as our backbone network. Combining with the generated dynamic prefix, we can represent a text of query as follows:

$$\boldsymbol{v}_q = \mathrm{MaxPooling}(\log\big(1 + \mathbf{ReLU}(\mathbf{LM}(\mathbf{PL}, \mathbf{X}; \theta_q)))) \in \mathbb{R}^d. \tag{5}$$

where $\mathbf{ReLU}$ is the Rectified Linear Unit. Following the common practice, we adopt the contrastive loss (Karpukhin et al., 2020; Khattab & Zaharia, 2020; Formal et al., 2021b) that utilize a limited number of positive documents $d_+$ and $d_-$ for each queries for training. Specifically, we employ BM25[3] to retrieve top-100 relevant documents as $d_-$ as negative samples except those also contain answers to a query. To encode positive documents and negative documents into corresponding vector representation $\boldsymbol{v}_{d+}$ and $\boldsymbol{v}_{d-}$, we employ a document encoder $\theta_d$ to encode them but skip the prefix generation as $\boldsymbol{v}_d = \mathrm{MaxPooling}(\log\big(1 + \mathbf{ReLU}(\mathbf{LM}(d; \theta_d)))) \in \mathbb{R}^d$.[4] Thus, a likelihood distribution can be formatted as follows,

$$P(d_+|\boldsymbol{v}_q, d_-; \theta_q, \theta_d) = \frac{\exp(\boldsymbol{v}_q^{\mathrm{T}} \boldsymbol{v}_{d+})}{\sum_{d' \in d_+ \cup d_-} \exp(\boldsymbol{v}_q^{\mathrm{T}} \boldsymbol{v}_{d'})}. \tag{6}$$

---

[2]We also report evaluation results using the dense model (e.g., DPR) as the backbone network in Table 1.

[3]We adopt the default setting of Anserini for BM25 where $k1 = 0.9, b = 0.4$.

[4]Using dynamic representations of different documents requires to rebuild index in real-time which causes heavy calculation cost.

Following the common practice of training lexical retriever (Formal et al., 2021b), we add additional floating point operations per second (FLOPS) regularization terms (Paria et al., 2020) to reduce the computation cost of the retrieval process and improve the retrieval effectiveness. Then, the loss function of our methods can be defined as

$$\mathcal{L}_q(\mathcal{C}) = \sum_{q \in \mathcal{C}} P(d_+ | \boldsymbol{v}_q, d_-; \theta_q, \theta_d) + \lambda_q \text{FLOPS}(\boldsymbol{v}_q) + \lambda_d \text{FLOPS}(\boldsymbol{v}_d), \tag{7}$$

where $\text{FLOPS}(\cdot)$ is a regularization term proposed by (Formal et al., 2021b) and we use hyper-parameters $\lambda_q, \lambda_d$ to adjust the sparsity of representation of $\boldsymbol{v}_q$ and $\boldsymbol{v}_d$, respectively.

**Model Training**    In training, we adopt a multi-tasking training paradigm with a mini-batch mixup which means that we randomly sample from all tasks to compose a training batch. We train the entire model on KILT for in-domain testing for superior performance. For cross-domain zero-shot retrieval, we freeze the parameters of the backbone network and only tune the parameters of our proposed components. The objective function we used can be described as:

$$\min_{\substack{\mathbf{P}, \theta_I, \theta_p, \\ (\theta_q, \theta_d)}} \mathcal{L}_q(\mathcal{C}) + \lambda_c \mathcal{L}_{\text{CPR}}(\mathcal{C}), \tag{8}$$

where $\mathcal{C} = \bigcup_{i=1}^{t} \mathcal{T}_i$ is the mixed data of different tasks, $\lambda_c$ is a fixed pre-defined weight to control the regularization of CPR.

## 3    EXPERIMENTS

**Benchmark Datasets**    We use two publicly available retrieval datasets for evaluation, including KILT (Petroni et al., 2021) and BEIR (Thakur et al., 2021). KILT is a benchmark for knowledge-intensive tasks that require retrieving additional knowledge from the wiki, we select all 7 datasets containing corresponding training samples to train our model in a multi-task setting. Specifically, we use a variety of tasks including fact-checking (FEVER), entity linking (AY2), slot filling (zsRE), question answering (NQ, TQA, HoPo), and dialogue (WoW). KILT provides the provenances of all tasks in one wiki corpus, which enables us to train models with a share passage encoder (Maillard et al., 2021). For BEIR, it is a widely known zero-shot information retrieval benchmark and we employ it to evaluate the transfer learning ability provided by different methods. Also, we remove the datasets that are contained in KILT which results in 10 tasks including TREC-COVID (TC), NFCorpus (NFC), ArguAna (Argu), Tóuche-2020 (Touche), FiQA-2018 (FiQA), DBPedia (DBP), SCIDOCS (SciD), Climate-FEVER (CFever) for a fair comparison.

**Evaluation Metrics**    When evaluation on KILT, we adopt R-Precision as the retrieval metric which is the primary metric used in their paper (Petroni et al., 2021). R-Precision can be calculated as $\frac{r}{R}$, where $R$ is the number of Wikipedia pages inside each provenance set and $r$ is the number of relevant pages among the top-R retrieved pages. For experiments on BEIR, Normalised Cumulative Discount Gain (nDCG) (Gysel & de Rijke, 2018) is reported to represent performances of different methods. Specifically, we utilize the official TREC evaluation tool (Gysel & de Rijke, 2018) and compute nDCG@10 for all datasets.

**Experiment Setups**    We train our model on KILT in a multi-task learning paradigm. Since our method can be combined with any transformer-based neural retriever, we adopt SPLADEv2 (Formal et al., 2021b) and DRP (Karpukhin et al., 2020) provided by the original paper as our backbone model[5]. The learning rate of the backbone network and the modules of HYPER is set to $2 \times 10^{-5}$ by following Formal et al. (2021b) and $1 \times 10^{-3}$ selected from $\{10^{-1}, 10^{-2}, 10^{-3}\}$, respectively. The temperature $\tau$ is set to $e$, $\lambda_q$ is set to 0.3, $\lambda_d$ is set to 0.1. $d_q$ is set to 400 and $d_p$ is set to 100. The number of train epochs is set up to 10 epochs, both max document length and max query length are set to 512 to fit the task with a very long query, and the batch size is set to 256. For each query, we provide 1 positive sample and 19 negative samples for training. We set the sequence length of each basic prompt to 100 selected from $\{10, 50, 100\}$. The $\lambda_c$ and the number of shared basic prompts $N$

---

[5]Both models are pre-trained on MS-MARCO (Nguyen et al., 2016) and can provide superior initial performance on KILT.

Table 1: Page-level R-precision on KILT. All models in the multi-tasking part are trained on 7 tasks of KILT, while the models in the zero-shot part are trained with the leave-one-out setting that leaves out the dataset used for testing in training. Model names that end with "zero" mean they are tested directly without training and the "-prefix" follows the model name means the corresponding model is trained through prefix-tuning. * indicates results from (Maillard et al., 2021).

| Model | FEVER | AY2 | zsRE | NQ | HoPo | TQA | WoW | AVG |
|---|---|---|---|---|---|---|---|---|
| Multi-Tasking | | | | | | | | |
| DPR* (Karpukhin et al., 2020) | 75.35 | 28.45 | 81.49 | 58.53 | 41.95 | 60.39 | 43.52 | 55.66 |
| DPR-prefix | 75.71 | 30.32 | 81.24 | 59.42 | 42.12 | 61.67 | 46.16 | 56.65 |
| DPR-HYPER | 76.35 | 31.80 | 81.79 | 60.83 | 42.67 | 62.91 | 46.35 | 57.53 |
| SPLADE (Formal et al., 2021a) | 81.23 | 34.29 | **84.59** | 58.17 | 50.29 | 60.30 | 47.38 | 59.46 |
| SPLADE-prefix | 81.56 | 35.83 | 83.98 | 59.25 | 50.63 | 61.34 | **50.13** | 60.38 |
| SPLADE-HYPER | **82.17** | **38.52** | 84.23 | **60.41** | **51.24** | **62.76** | 49.84 | **61.31** |
| Zero-Shot | | | | | | | | |
| BM25* | 50.13 | 3.47 | 66.43 | 25.83 | 43.95 | 29.44 | 27.50 | 35.25 |
| DPR-zero (Karpukhin et al., 2020) | 54.66 | 2.03 | 34.69 | 53.22 | 21.95 | 45.01 | 27.07 | 34.08 |
| DPR (Karpukhin et al., 2020) | 54.75 | 1.62 | 36.65 | 51.62 | 13.14 | 48.25 | 24.24 | 32.90 |
| DPR-prefix | 56.49 | 2.65 | 36.31 | 51.53 | 18.34 | 50.27 | 27.06 | 34.66 |
| DPR-HYPER | 59.13 | 3.56 | 37.03 | 53.31 | 19.13 | 52.24 | 29.06 | 36.21 |
| SPLADE-zero (Formal et al., 2021a) | 72.05 | 2.90 | 76.77 | 49.66 | 47.65 | 48.12 | 43.35 | 48.64 |
| SPLADE (Formal et al., 2021a) | 73.42 | 2.32 | 80.75 | 47.78 | 29.96 | 51.63 | 38.73 | 46.37 |
| SPLADE-prefix | 75.12 | 3.21 | 81.24 | 48.36 | 40.25 | 54.42 | 43.55 | 49.45 |
| SPLADE-HYPER | **78.39** | **5.08** | **82.16** | **50.02** | 41.38 | **56.28** | **46.81** | **51.44** |

are tuned and we finally select 0.1 and 20 respectively[6]. We fix the random seed always to 42 and all experiments are conducted on eight A100 GPUs. [7]

## 3.1 MAIN EVALUATION

**Supervised and Zero-shot Performance on KILT**   We conduct both supervised and zero-shot experiments on KILT and the results are shown in Table 1. Since our hyper-prompted training mechanism is applied to the lexicon-weighted retrieval method (e.g., SPLADE), we name it as SPLADE-HYPER. Besides, we also test the effectiveness of our HYPER on dense retrieval methods (e.g., DPR) which results in DPR-HYPER. First, we can easily find that SPLADE can provide superior performance than dense representation methods (e.g., DPR) in terms of both the supervised and zero-shot settings on KILT. Notably, the performance gap is extremely significant in the zero-shot setting where dense retrieval methods (a.k.a., DPR) can only achieve comparable results with traditional BM25[8] while lexicon-weighted retrieval methods significant outperform BM25 and dense retrieval methods. Second, compared with fine-tuning model entirely or prefix-tuning (Liu et al., 2021b), our SPLADE-HYPER can obtain better performance on most tasks of the supervised setting, which demonstrates the superiority of our HYPER in sharing and transferring knowledge across different retrieval tasks or domains. Meanwhile, we can notice that HYPER can also improve performance even in all unseen tasks, which reflects our method can effectively transfer learned knowledge from previous tasks and adapt to different types of queries across different domains. The experimental results of the few-shot setting (shown in Appedix A) can also further prove the effectiveness of the proposed HYPER.

**Zero-shot Performance on BEIR**   We also directly test the performance of our SPLADE-HYPER without tuning on BEIR, which is a widely known zero-shot IR benchmark. Following the most recent works (Xu et al., 2022; Wang et al., 2022), we compare our methods with varieties of methods, including DocT5 (Nogueira et al., 2019a), ColBERT (Khattab & Zaharia, 2020), DPR (Karpukhin et al., 2020), ANCE (Xiong et al., 2020), GenQ (Thakur et al., 2021), TAS-B (Reimers & Gurevych, 2019), MoDIR (Xin et al., 2021) and LapraDOR (Xu et al., 2022). Experiment results are shown in Table 2, as we can see, our HYPER occupies the best performance on 4 of 9 tasks of BEIR, and we also attain the best average performance. Besides, our method consistently outperforms the backbone

---

[6]The effects of different hyperparameters are investigated in section 3.2

[7]Our Code is available at `https://github.com/oklen/HypeR`.

[8]The observation is consistent with several previous studies (Maillard et al., 2021; Thakur et al., 2021)

Table 2: Zero-shot generalization evaluated on 9 datasets of BEIR. * indicates results from Thakur et al. (2021). † indicates results from Xu et al. (2022).

| Model | TC | NFC | FiQA | Argu | Touche | DBP | SciD | CFever | SciF | AVG |
|---|---|---|---|---|---|---|---|---|---|---|
| BM25* | 65.6 | 32.5 | 23.6 | 31.5 | **36.7** | 31.3 | 15.8 | 21.3 | 66.5 | 38.8 |
| BM25+CE* | 75.7 | 35.0 | 34.7 | 31.1 | 27.1 | 40.9 | 16.6 | 25.3 | 68.8 | 39.5 |
| DocT5* (Nogueira et al., 2019a) | 71.3 | 32.8 | 29.1 | 34.9 | 34.7 | 33.1 | 16.2 | 20.1 | 67.5 | 37.3 |
| ColBERT† (Khattab & Zaharia, 2020) | 67.7 | 30.5 | 31.7 | 23.2 | 20.2 | 39.2 | 14.5 | 18.4 | 67.1 | 34.7 |
| DPR† (Karpukhin et al., 2020) | 33.2 | 18.9 | 11.2 | 17.5 | 13.1 | 26.3 | 7.7 | 14.8 | 31.8 | 19.4 |
| ANCE† (Xiong et al., 2020) | 65.4 | 23.7 | 29.5 | 41.5 | 24.0 | 28.1 | 12.2 | 19.8 | 50.7 | 32.8 |
| GenQ† (Thakur et al., 2021) | 61.9 | 31.9 | 30.8 | 49.3 | 18.2 | 32.8 | 14.3 | 17.5 | 64.4 | 35.7 |
| TAS-B† (Reimers & Gurevych, 2019) | 48.1 | 31.9 | 30.0 | 42.9 | 16.2 | 38.4 | 14.9 | 22.8 | 64.3 | 34.4 |
| MoDIR (Xin et al., 2021) | 67.6 | 24.4 | 29.6 | 41.8 | 31.5 | 28.4 | 12.4 | 20.6 | 50.2 | 34.1 |
| LapraDOR† (Xu et al., 2022) | 72.8 | 34.6 | 31.7 | **50.7** | 32.2 | 36.1 | **18.5** | 22.8 | 69.7 | 41.0 |
| SPLADE | 57.4 | **42.3** | 23.0 | 26.6 | 21.3 | 38.1 | 13.9 | 23.6 | 64.6 | 34.5 |
| SPLADE-Prefix | 73.0 | 32.9 | 34.7 | 49.3 | 25.2 | 41.8 | 15.3 | 22.3 | **70.5** | 40.6 |
| SPLADE-HYPER | **79.1** | 33.6 | **35.1** | 50.1 | 27.4 | **43.6** | 15.6 | **23.7** | 70.4 | **42.1** |

Table 3: Ablation study on KILT.

| Model | FEVER | AY2 | zsRE | NQ | HoPo | TQA | WoW | AVG |
|---|---|---|---|---|---|---|---|---|
| SPLADE-HYPER | **82.17** | **38.52** | **84.23** | **60.41** | **51.24** | **62.76** | 49.84 | **61.31** |
| SPLADE-HYPER w/o query-conditional | 80.61 | 36.65 | 83.24 | 59.07 | 50.60 | 61.37 | 49.48 | 60.15 |
| SPLADE-HYPER w/o alignment $\mathcal{D}_f$ | 80.82 | 37.47 | 83.97 | 59.18 | 50.19 | 61.54 | 50.57 | 60.53 |
| SPLADE-HYPER w/o uniformity $\mathcal{D}_f$ | 81.39 | 36.34 | 83.67 | 59.42 | 50.53 | 61.52 | **50.95** | 60.55 |
| SPLADE-prefix | 81.56 | 35.83 | 83.98 | 59.25 | 50.63 | 61.34 | 50.13 | 60.38 |

network SPLADE, which indicates HYPER can enable a model to transfer learned knowledge across different domains better and thus improve the generalization ability of models. Moreover, our SPLADE-HYPER is better than SPLADE-prefix, which shows the superiority of proposed query-conditional prompt synthesizing and the strong ability of the dynamic parameterization to adapt different types of queries across the different domains.

## 3.2 ANALYSES

**Ablation Study**    To verify the effectiveness of our proposed mechanisms, we propose several variants of our model for comparison. Specifically, to evaluate the effectiveness of QPS, we replace the generated attention score distribution with a fixed uniform distribution which results in a variant of our model without the attention module. Also, to verify the effectiveness of CPR, we separately remove the alignment regularization and uniformity regularization to constitute the other two variants. Experiment results are shown in Table 3, as we can see, removing any modules in our mechanisms cause a significant decrease in performance. Comparison between HYPER and HYPER w/o query conditional indicates the attention module in QPS can successfully generate suitable prompts for different queries and thus improve the performance. Moreover, we can observe removing either regularization terms of CPR results in degraded performances. Therefore, we can conclude that both alignment regularization and uniformity are crucial for enabling the effective query-conditional prompt generation to process different queries.

**Prompt Attention Similarity vs Task Similarity**    We conduct additional experiments to investigate whether the similarities of prompt attention distributions can reflect the similarities between different tasks. Specifically, we calculate the similarities between the mean values of attention score distributions $\mathcal{A}$ belonging to the same task, and the results are shown in Figure 2. Obviously, there are two different groups of similar attention score distribution, AY2 and WoW in the top-left corner and others in the bottom-right, which are bounded by green lines. After reviewing the data, we find the two groups of datasets can be distinguished by the lengths of input queries.

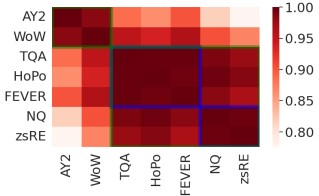

Figure 2: Similarities of attention scores distributions of different tasks.

In particular, the lengths of queries of AY2 and WoW are usually composed of multiple sentences while queries of other datasets only contain one or few sentences. This implies our mechanism can recognize the different lengths of different queries, which is a prerequisite for dynamically adopting

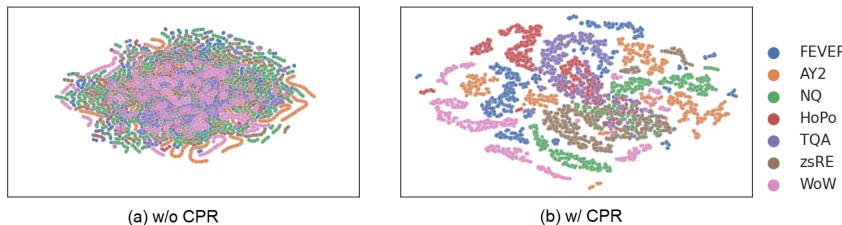

(a) w/o CPR

(b) w/ CPR

Figure 3: Visualization of attention weight embeddings for different tasks.

different methods to process different queries, such as extracting important information from long queries or predicting implicit relations in short queries. Moreover, we take a closer look and observe the sub-groups of attention score distributions annotated by blue lines. Comparing the queries of these two groups, we find the group composed of TQA, HoPo, and FEVER requires more inference skills than the group composed of zsRE and NQ. This implies our query-conditional module can even distinguish more subtle differences in queries, which qualitatively reflects the effectiveness of QPS.

**Visualization of Prompt Attention Distribution**  To further confirm that CPR works as our expectation that prompts the distributions of attention scores to be uniform and aligned, we employ dimension reduction such as t-SNE to visualize them. To this end, we randomly draw the 2000 samples from the test split of each dataset and then employ t-SNE to transform normalized prompt attention (a.k.a, $\mathcal{A}$) into 2D vectors. Experiment results are shown in Figure 3. The comparison between w/ CPR and w/o CPR indicates that (1) uniformity regularization isolates attention score distribution of dissimilar queries belonging to different tasks, which helps avoid mode collapse. (2) alignment regularization prompts the attention score distributions of similar queries belonging to the same tasks to become closer to each other, which may improve the effectiveness of training prefix merged by specific pattern attention scores and thus benefits the model performance.

**Performance vs Efficiency**  We further investigate the efficiency of our method since real-world applications not only require better retrieval results but also low retrieval latency. To measure the latency, we adopt the Query Per Second (QPS) as a metric and the higher QPS means more queries can be processed in time. We evaluate both dense retrievers and sparse retrievers with or without HYPER on KILT and experiment results are shown in Figure 4. As we can see, compared with backbone models, HYPER can obtain better performance and higher QPS which demonstrates the better efficiency of our method. Meanwhile, although BM25 obtains better efficacy, neural methods outperform it on the retrieval metric significantly.

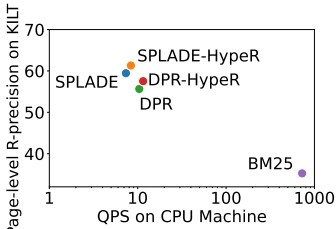

Figure 4: Performance versus QPS (latency).

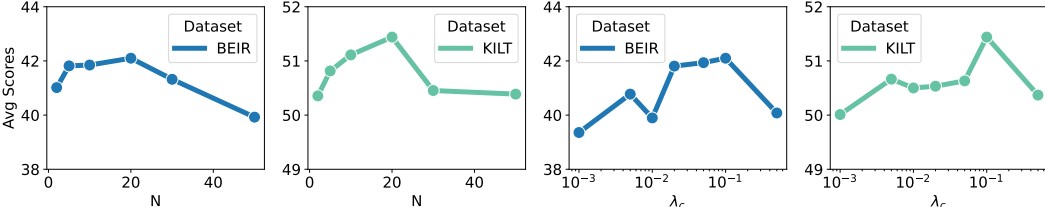

Figure 5: Effect of the number of shared basic prompts ($N$) and weights of CPR ($\lambda_c$). The two figures on the left vary $N$, while the two on the right vary $\lambda_c$.

**Impact of Parameters**  To better understand the effect of our method, we change the number of basic prompts ($N$) and the weight of CPR ($\lambda_c$) and test these variants on both KILT and BEIR. Specifically, we vary the number of shared prompts in $\{2, 5, 10, 20, 30, 50\}$ with the weight of CPR fixed to 0.1. Meanwhile, we vary the weights of CPR in $\{0.001, 0.005, 0.01, 0.02, 0.05, 0.1, 0.5\}$ with number of shared prompt is fixed to 20. The average scores of different variants are shown in Figure 5. As the number of shared prompts varies across different values, we can observe the performance increase and then decrease. This phenomenon implies more shared prompts can enable the model to learn more patterns exits in data while too many shared prompts suffered from the sparsity of representation space and thus results in insufficiently trained combinations of

shared prompts. Similarly, we also find a moderate weight of CPR can lead to the best performance. This indicates our proposed CPR can benefit the query conditional module and thus improve the performance while too large $\lambda_c$ causes the main objective loss $\mathcal{L}_q$ to be ignored.

## 4 RELATED WORK

Information retrieval can be generally defined as searching relevant documents about a short text as an input query. The major challenge of IR comes from the huge amount of candidate documents, which results in the matching function between query and documents having to be extremely simple for high efficacy. Therefore, we need to seek superior encoding methods for queries and documents to improve the accuracy of retrieval. Traditional methods such as tf-idf and BM25 rely on term matching and build high-dimension, sparse vector (Yang et al., 2017; Robertson & Zaragoza, 2009) for effective retrieval. Although they are effective across various tasks of different domains (Chen et al., 2017; Yang et al., 2017), they fail to adapt to more specific tasks and are outperformed largely by the recently popular neural text retriever with sufficient training samples.

Neural text retrievers are based on pre-trained language models such as BERT (Devlin et al., 2019) and can be classified into two types, the dense-vector retrievers (Xiong et al., 2021) and sparse-vector retrievers. For dense-vector retriever (Karpukhin et al., 2020; Gao & Callan, 2021), it encodes both queries and documents into low-dimension vectors and then calculates the relevance scores between them. Although a dense vector is more effective to conduct semantic retrieve (Lin & Lin, 2022), the compressing process of texts may result in a lost of information. In the contrast, sparse-vector retriever (Formal et al., 2021b; Khattab & Zaharia, 2020) encodes both queries and documents into high-dimension, sparse vectors and then calculate the concurrence (Nogueira et al., 2019a) or top coordinate terms (Formal et al., 2021b) of words. Therefore, it can achieve effective lexicon matching, and the varying amount of activating dimensions in vectors relieves the information lost in encoding, while it introduces an additional quantization process to avoid the unstable of real-values of vectors.

Although neural retrievers can perform well with a moderate amount of data, in a real-world application, the data of target tasks could be considerably scarce (Thakur et al., 2021). Hence, zero-shot and few-shot settings on retrieval tasks receive more and more attention and various methods have been proposed to improve the model performance under this setting including unsupervised pretrained (Xu et al., 2022; Wang et al., 2022), data augmentation (Nogueira et al., 2019b; Thakur et al., 2021; Dai et al., 2022) and enhanced training (Reimers & Gurevych, 2019; Xiong et al., 2020). However, there is only a primary investigation of methods utilizing transfer learning, and still a large room for improvement. Technically, HYPER is similar to utilizing prefix tuning for IR tasks as Tam et al. (2022) and discrete prompt tuning for natural language tasks as Sanh et al. (2022), but goes beyond the comparison of existing mechanisms and focuses on generating dynamically query-conditional prompts, and enabling a neural retriever to process queries of different tasks uniformly. Besides, HYPER is inspired by HyperPrompt (He et al., 2022) that explores prompt-based task-conditioning of self-attention in Transformers. Nonetheless, HYPER dynamically generates the prompt according to every query itself rather than indispensably relying on the task-level information, which enables our model to obtain superior generalization and transferability. Moreover, HYPER employs the prefix-tuning method to utilize the dynamic prompts rather than concatenating prompts into the self-attention layer directly.

## 5 CONCLUSION

In this paper, we propose to process queries of different tasks uniformly regardless of the difference in query format and varying priorities of query vectors. Specifically, we present HYPER, a hyper-prompted training mechanism that can be easily combined with any transformer-based neural retrievers. In HYPER, to enable the uniform process queries, we propose Query-conditional Prompt Synthesizing (QPS) to dynamically synthesize different parameter combinations for different queries. Moreover, to resolve the mode collapse problem of attention scores distribution in QPS, we propose Contrastive Prompt Regularization (CPR) to simultaneously learn representations of hyper-prompts and clustering for different types of queries across different domains or tasks. We conduct extensive experiments which demonstrate our methods can improve the in-domain and out-of-domain zero-shot retrieval performance of a neural retriever significantly. In-depth analyses reveal how our mechanism enables the uniform processing of queries.

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

## A  FEW-SHOT EVALUATION ON KILT

In Table 4, we continually train models in a new task with a few-shot setting. As we can see, our methods consistently outperform several strong baselines, which demonstrates that our method benefits the model more quickly adapting to the different tasks in the same domain by sharing knowledge among tasks.

Table 4: Page-level R-precision on KILT in the few-shot setting.

| Target Data | NQ | | | | | TQA | | | | |
|---|---|---|---|---|---|---|---|---|---|---|
| #Instance | 192 | 384 | 576 | 1152 | 2304 | 192 | 384 | 576 | 1152 | 2304 |
| SPLADE | 48.19 | 48.32 | 48.55 | 48.97 | 49.56 | 51.97 | 51.32 | 51.66 | 53.53 | 53.18 |
| SPLADE-prefix | 49.07 | 49.40 | 50.10 | 50.51 | 50.42 | 54.62 | 54.01 | 53.37 | 55.48 | 56.23 |
| SPLADE-HYPER | **50.27** | **50.24** | **51.13** | **51.43** | **51.96** | **56.56** | **55.81** | **55.07** | **57.68** | **58.24** |
| Target Data | FEVER | | | | | zsRE | | | | |
| #Instance | 192 | 384 | 576 | 1152 | 2304 | 192 | 384 | 576 | 1152 | 2304 |
| SPLADE | 73.42 | 73.61 | 73.75 | 74.01 | 74.72 | 80.83 | 81.13 | 81.24 | 81.45 | 81.78 |
| SPLADE-prefix | 75.12 | 75.19 | 74.86 | 75.74 | 76.31 | 81.30 | 81.72 | 81.65 | 82.09 | 82.36 |
| SPLADE-HYPER | **78.37** | **78.44** | **77.94** | **78.55** | **79.08** | **82.19** | **82.58** | **82.63** | **82.69** | **83.44** |

## B  COMPARISON OF THE TASK-SPECIFIC FINE-TUNING MODEL

We also test the performance of directly fine-tuning the SPLADE on each task of KILT, which results in 7 different retrievers (SPLADE-FT). The results are shown in Table 5. We can find that SPLADE-FT can achieve a significantly better AVG score than SPLADE-MT, although multi-task training can bring improvement to 4 out of 7 tasks (a.k.a. FEVER, zsRE, NQ, and TQA). Besides, through incorporating the proposed HYPER, SPLADE-HypeR can achieve a comparable AVG score with SPLADE-FT and even outperform SPLADE-FT on 4 out of 7 tasks (a.k.a. FEVER, zsRE, NQ, TQA, and WoW). Notably, SPLADE-FT train separate models for each task which results in roughly 6 times more parameters than our model.

Table 5: Comparison of the task-specific fine-tuning model on KILT. SPLADE-prefix means the model trained through prefix-tuning.

| Model | FEVER | AY2 | zsRE | NQ | HoPo | TQA | WoW | AVG |
|---|---|---|---|---|---|---|---|---|
| SPLADE-FT | 80.20 | 55.02 | 83.97 | 57.00 | 51.29 | 55.40 | 48.69 | 61.65 |
| SPLADE-MT | 81.23 | 34.29 | 84.59 | 58.17 | 50.29 | 60.30 | 47.38 | 59.46 |
| SPLADE-prefix | 81.56 | 35.83 | 83.98 | 59.25 | 50.63 | 61.34 | 50.13 | 60.38 |
| SPLADE-HYPER | 82.17 | 38.52 | 84.23 | 60.41 | 51.24 | 62.76 | 49.84 | 61.31 |

## C  EFFICIENCY OF QUERY ENCODING

We further investigate how the length of prefixes influences the computation cost of query encoding and retrieval performance. To this end, we vary the size of the prefix in {5,10,20,50,100, 200} and record both the average time of encoding a query and retrieval performance on KILT. We compare our method with Prefix-tuning and the results are shown in Figure 6. The inference time is measured on a machine with Intel Xeon CPU E5-2678. As we can see, our HypeR obtains better average retrieval scores than Prefix-Tuning for all different prefix lengths, and both Prefix-Tuning and HypeR achieve the best performance when the prefix length is 100. Notably, our HypeR costs 4% and 25% more encoding time than Prefix-Tuning when the prefix length is 10 and 100, respectively. We have put the results as a figure in the appendix of our revised manuscript. Thank you again for your constructive suggestions.

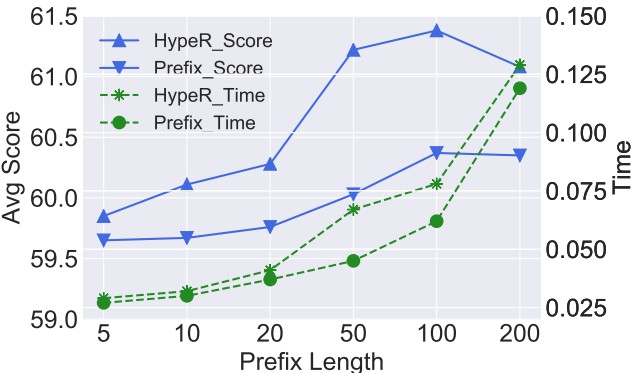

Figure 6: Efficiency compraing between HypeR and Prefix-Tuning. The multi-tasking fine-tuning can be viewed as an origin.

## D    CASE STUDY

To provide an intuitive understanding of our methods, we show some similar and dissimilar queries based on prompt attention distributions in Table 6. Specifically, we randomly sample 3 queries belonging to different tasks as base queries from the collection of test data of all KILT datasets. Then, for each base query, we randomly sample 2 queries from the queries with top 5% similar attention scores. Besides, we also randomly sample 2 queries belonging to different datasets from the queries with the lowest 5% similar attention score, since we find there are too many trivial candidates from the same datasets.

Table 6: Queries with similar and dissimilar prompt attention distributions.

| Query Type | Source Task | Query Content |
|---|---|---|
| *Case 1* | | |
| Base Query | Hotpotqa | Which film was made more recently, The Diplomat or Rien que les heures? |
| Similar Query 1 | Hotpotqa | Which movie was released more recently, Waking Sleeping Beauty or Mars Needs Moms? |
| Similar Query 2 | Hotpotqa | Who directed a film more recently, Don Bluth or Raoul Walsh? |
| Dissimilar Query 1 | Aidayago2 | Action Performance to acquire firms .  [START_ENT] TEMPE [END_ENT], Ariz. 1996-12-06 Action Performance Cos Inc said Friday it has agreed to acquire Motorsport Traditions Ltd and Creative Marketing & Promotions Inc for about $13 million in cash and stock. The two firms to be acquired have about $25 million in annual revenues from the design, manufacture and sale and distribution of licensed motorsports products. The deal is expected to close by the end of the year subject to due diligence and other customary closing conditions. |
| Dissimilar Query 2 | WoW | I love American Football, the first game was played between Rutgers and Princeton in 1869. Wow I had no idea it was that recently played! Yep, it can actually be traced to early versions of rugby football also. Who is your favorite team. Probably the New England PAtriots. I always had a thing for Tom Brady lol. Did he not deflat footballs or something one year ? |
| *Case 2* | | |
| Base Query | WoW | Do you shop online for clothes? Yes quite often. It allows me to directly buy goods from a seller over the internet without having to leave the comfort of my home.  What about you?  Yeah same I love visiting websites of different retailers directly to see product availability and the best prices. Yeah me too. I use amazon a lot to buy stuff. Like even this computer! |

| | | |
|---|---|---|
| Similar Query 1 | WoW | Online shopping is the better experience to choose the the product on our desire.Yes! I much prefer shopping online at home on the Internet rather than fighting crowds in an actual store. what did you often purchase? Anything and everything! Housewares, clothing, electronics, you name it! I like the convenience of browsing stores on my laptop, tablet computer and smartphone! yes, it includes lot of choices I especially like the functionality of websites like Amazon for shopping. The have a great search feature that allows me to find specific models, brands and items. Such website, improved a lot for customer convenience. |
| Similar Query 2 | WoW | I like to shop online, probably a bit too much. Lol me too! What's your favorite online store? I love to shop on amazon. So you shop online a lot too? Where do you shop? Amazon as well. I also shop a lot online at Old Navy and Gap. They have great sales. Do they really? I mainly use only amazon. Do you use Amazon Kindle, the series of e-readers designed and marketed by Amazon. No I have never used a kindle. My dad has one of those though and it looks cool if you love books. |
| Dissimilar Query 1 | FEVER | Hrithik Roshan was a film star. |
| Dissimilar Query 2 | Hotpotqa | When did the theatre open that has Valery Abisalovich Gergiev as it's artistic director? |
| *Case 3* | | |
| Base Query | FEVER | Reliance Industries only works in textiles. |
| Similar Query 1 | FEVER | Sanjjanaa works in the Telegu film industry. |
| Similar Query 2 | FEVER | Rakul Preet Singh mostly works in the Telugu film industry. |
| Dissimilar Query 1 | WoW | Have you ever read anything by John Grisham? Yes, I have read his very first novel "A Time to Kill" which was published in June 1989 after he took four years to write it! I haven't read anything by him but I remember the movies for both a time to kill and the firm. |
| Dissimilar Query 2 | Aidayago2 | Multinational commander going back to east Zaire. Jonathan Wright NAIROBI 1996-12-06 The Canadian general in charge of a multinational force for eastern Zaire said on Friday he was going back to Zaire for more information about the plight of about 165,000 Rwandan refugees adrift in the countryside. Lieutenant-General Maurice Baril told a news conference in Nairobi his main concern was for a large group of about 150,000 refugees living off the land in a valley about 65 km (40 miles) west of the eastern city of Goma. If he decided it was necessary and safe for the aircrew, he would not hesitate to order airdrops of food for the refugees, even against the wishes of the government in Kinshasa and the [START_ENT] Zairean [END_ENT] rebels who control much of eastern Zaire, he said. " Tomorrow I ḿ going into Rwanda and my intention is to go across into eastern Zaire and try to find out for the second time what the situation is on the ground," he said. General Baril saw rebel leader Laurent Kabila in Goma last week but the rebels told him the crisis was over because most of the Rwandan refugees have already gone home. The rebels do not want the multinational force to deploy on the ground , for fear it might help the Zairean army regain control of the area. |

