# OpenReview forum: "HypeR: Multitask Hyper-Prompted Training Enables Large-Scale Retrieval Generalization"
_ICLR.cc/2023/Conference — ICLR 2023 poster_

### Official Review · Reviewer_1Ry7 · 2022-10-24

**Confidence:** 4
**Correctness:** 4
**Technical Novelty And Significance:** 3
**Empirical Novelty And Significance:** Not applicable
**Recommendation:** 8

**Clarity, Quality, Novelty And Reproducibility:**

Clarity: very clear, with few minor changes to improve on.

Quality: solid and sound with extensive experiment results to back up the claim.

Novelty: is novel in dynamically conditioning prompts on the query-level instead of task-level.

Reproducibility: The authors state that "Our implemented codes will be made public."


**Strength And Weaknesses:**

Pros:
- The paper presents an effective approach to address a practical problem of domain adaptation in retrieval, following on the recent popular direction of prompt tuning and prefix tuning.
- The approach is novel. Compared with previous related work (e.g. Hyper-Prompt), HYPER addresses retrieval task across different domains/datasets, so it dynamically generates/conditions distinct prompts on the query-level instead of task-level.
- The approaches, model architecture, formulas, and experiments are sound and solid, very clearly explained, and easy to follow. The paper is well-organized and written.
- Experiments are conducted on popular public datasets of KILT and BEIR, and are compared against the recent SOTA systems in the retrieval field. The results and conclusions are convincing.
- The discussions and analysis are detailed and insightful. The analysis around prompt attention similarity and task similarity, together with visualization, is interesting and informative.

Cons:
- The approach is employed on query encoder. It would be interesting to explore and discuss about the effect of transfer learning on the document encoder as well. Also, I am a little curious about employing HYPER in retrieval models where the query encoder and document encoder share a large part of parameters, for example, ColBERT.
- A small suggestion: It would be interesting to add some query examples (maybe in Appendix) with similar and dissimilar prompt attention scores.


Question:
Page 3, when describing how to obtain the query representation Q, should it be generalized to any retrieval model's query encoder? Or, maybe mention that this description is specific to a particular model, and the other models follow in a similar way?

Minor editing errors:
Page 7, 'we calculate the mean...' -> should it be 'we calculate the similarities between the mean...' ?
Page 9, 'considerably scare' -> 'considerably scarce'


**Summary Of The Paper:**

The paper proposes HYPER, a mechanism for transfer learning among retrieval tasks in different domains and datasets. For each query, it first dynamically generates a hyper-prompt to synthesize the query by attending to a given number of shared prompts, then encodes the synthesized prompt into prefixes, and employs the prefix-tuning method to train the retriever model in a multi-task setting. Contrastive learning is employed to cluster synthesized prompts based on the datasets they come from. Experiment results show the efficacy of HYPER, especially on 0-shot in-domain and cross-domain retrieval settings.


**Summary Of The Review:**

The paper addresses a well motivated problem effectively with recent advances in prefix and prompt tuning, and with innovations on conditioning prompts on the query-level instead of task-level. The claims are backed up by extensive experiments and analysis on recent public datasets compared with SOTA systems.

---

> ### Author Response · Authors · 2022-11-15
> **Response to Reviewer 1Ry7**
>
> We appreciate your constructive comments, and we have updated the manuscript accordingly.
>
> > **Question 1**: The approach is employed on query encoder. It would be interesting to explore and discuss about the effect of transfer learning on the document encoder as well. Also, I am a little curious about employing HYPER in retrieval models where the query encoder and document encoder share a large part of parameters, for example, ColBERT.
> - Thanks for this great question! Since we synthesize the prompt dynamically according to given queries, if we apply our method to the document encoder, we may update representations of large-scale collection of documents in runtime, which costs too much computation resources and hinders the fast retrieval during online inference. Alternatively, we can consider employing HYPER for both the query encoder and document encoder in some reranking settings, and we would like to leave it as future work. Thank you again for the great suggestion!
>
> > **Question 2**: It would be interesting to add some query examples (maybe in Appendix) with similar and dissimilar prompt attention scores.
> - Thanks for your constructive suggestions! We have put some cases in the appendix of our revision  (Appendix D). Please kindly refer to the updated version.
>
> > **Question 3**: Page 3, when describing how to obtain the query representation Q, should it be generalized to any retrieval model's query encoder? Or, maybe mention that this description is specific to a particular model, and the other models follow in a similar way?
> - The query representation Q is used for generating the dynamic prompt for the query encoder. After obtaining the dynamic prompt (PL), we compute the dynamic representation v_q in Equation (5) for retrieving, which is generalized to different retrieval models. Sorry for such a confusing description of Q. We have changed the description of Q as the incipient representation in the revision.
> - Our method and description can be generalized to any neural retrievers based on pre-trained language models. Notably, we also conducted experiments on two representative retrieval methods, including DPR and SPLADE.
>
> > **Question 4**: Page 7, 'we calculate the mean...' -> should it be 'we calculate the similarities between the mean...' ? Page 9, 'considerably scare' -> 'considerably scarce'
> - Thanks for pointing this out, and we've fixed the typos and grammatical errors in the revision.

---

### Official Review · Reviewer_BVqR · 2022-10-24

**Confidence:** 3
**Correctness:** 3
**Technical Novelty And Significance:** 3
**Empirical Novelty And Significance:** 3
**Recommendation:** 6

**Clarity, Quality, Novelty And Reproducibility:**

As mentioned above, the clarity and quality of this paper are good. The reproducibility is fine with clear experiment setups on page 5 and impacts of hyper-parameters on page 8.
However, the novelty part is a concern. Although the proposed method is more effective than prefix-tuning, it needs to answer a key question: is it the first method to construct **query-specific** prompt for different tasks? If not, how's the proposed method compared to previous methods?

**Strength And Weaknesses:**

Strength:
1. The problem of cross-domain text retrieval that this paper studies is important.
2. The paper is well written with clear motivation, method description, and experiment setup.
3. The empirical results are promising and comprehensive, covering two popular benchmarks KILT and BEIR, two representative retrieval methods DPR and SPLADE, and two settings supervised and zero-shot.

Weakness:
The main weakness of this paper is the novelty of the proposed HYPER mechanism. It lacks the comparison with other transfer learning mechanisms such as the ones mentioned in [1] and [2]. Although those papers do not conduct experiments on retrieval, it's easy to extend them to the retrieval setting. It seems to me that HYPER is also very general and can be applied to other downstream settings like GLUE. Thus it's necessary to have a discussion with other transfer learning mechanisms.

[1] He, Junxian, et al. "Towards a unified view of parameter-efficient transfer learning." arXiv preprint arXiv:2110.04366 (2021).
[2] Asai, Akari, et al. "Attentional Mixtures of Soft Prompt Tuning for Parameter-efficient Multi-task Knowledge Sharing." arXiv preprint arXiv:2205.11961 (2022).




**Summary Of The Paper:**

This paper proposes a hyper-prompted training mechanism for large-scale text retrieval across tasks of different domains.

**Summary Of The Review:**

Overall I found this paper clearly written with strong motivation and has extensive and promising experiment results, but the novelty contribution is unclear. Thus I think this paper as marginally above the acceptance threshold and highly recommend the authors to add the discussion and necessary baselines into this paper.

---

> ### Author Response · Authors · 2022-11-15
> **Response to Reviewer BVqR**
>
> Thanks for your insightful comments! We will try our best to address your concerns and answer your questions.
>
> > **Question 1**: It lacks the comparison with other transfer learning mechanisms such as the ones mentioned in [1] and [2].
> - We appreciate for pointing the two references out. Although both works are latest prompt-based transfer learning mechanisms, there are essential differences in task formation, model specification and applicable settings with the two references.
> - The first reference [1] provided a unified view of some basic transfer learning mechanisms including prefix-tuning, adaptor, and LoRA, and further introduced a scaled parallel adapter. However, all those transfer learning mechanisms are query independent, while we propose a prompt synthesizer that generates query-conditional prompts dynamically. Moreover, we focus on achieving better utilization of multi-task learning rather than designing a learning mechanism to improve the model performance on the given task. To more accurately investigate the improvement of model performance for multi-task learning, we adopt a simple but effective prefix as our basic component to utilize the query-conditional prompt. We also have a complete comparison of similar methods such as traditional prefixes and prompts. Alternatively, we can also adopt the methods in [1] to make use of the query-conditional prompt, for example placing it as a scaled parallel adapter. Since designing a more delicate mechanism to make use of the dynamically generated parameters is not our main objective, we would like to leave this part as future work to further improve our model.
>
> -  Although our approach shares high-level inspiration with concurrent [2] for dynamic soft prompts,  there are essential differences in model specification, leading to significantly different applicabilities. First, our proposed QPS combined with CPR implicitly normalizes the relationship between tasks and corresponding prompts which enables us to share prompts across different tasks, while [2] has to exactly provide a prompt as unique parameters for each task.  Second, our proposed CPR can also promote the learning of representations of hyper-prompts and clustering for different types of queries across different domains or tasks, which encourage similar tasks to benefit from each other. In contrast, [2] only provide tasks in the target area with knowledge from pretraining while the pretraining task themselves do not have a beneficial impact on each other. Last but most importantly, [2] has two training stages that first pre-train several prompts for each task to build a prompt pool and then train an attention module with fixed prompts of pretraining tasks on the target area. Therefore, it has to train the attention module on the target area and can hardly perform zero-shot directly in our setting.
> - Given task distinction, methodology divergence, applicability difference, and working concurrence, these references may not diminish the contributions of this work. Hope the above clarifications may address your concern.  We will also include the references and the above discussion in our revision.
>
>
> > **Question 2**: Although those papers do not conduct experiments on retrieval, it's easy to extend them to the retrieval setting. It seems to me that HYPER is also very general and can be applied to other downstream settings like GLUE.
> - Thanks for your great suggestions. Currently, our paper focus on large-scale retrieval, where the involved tasks share the same task formation and retrieval source, and thus different tasks can more easily benefit from each other. However, GLUE contains the single-sentence and pair-wise sentence semantic understanding tasks.  Moreover, there are large differences between different tasks which cause difficulty in sharing knowledge. Therefore, we do not conduct the experiment on GLUE, however, we are glad to extend the applicability of our work in the future. Thanks for your insightful suggestions again.
>
> > **Question 3**: Is it the first method to construct query-specific prompt for different tasks? If not, how's the proposed method compared to previous methods?
> - No. As discussed in the first question, the second reference [2] also introduces the query-specific prompt. Although our approach shares high-level inspiration with [2] for dynamic soft prompts, there are essential differences in model specification, leading to different application settings. Please kindly refer to our response for **Question 1**. Thank you again for pointing the important reference out.

---

### Official Review · Reviewer_tLxh · 2022-10-26

**Confidence:** 3
**Correctness:** 3
**Technical Novelty And Significance:** 3
**Empirical Novelty And Significance:** 3
**Recommendation:** 6

**Clarity, Quality, Novelty And Reproducibility:**

- Clarity: This work is generally well-written and easy to follow.
- Quality: This work is technically sound.
- Novelty: This work is somewhat novel.
- Reproducibility: No source code is shared, it might be challenging to reproduce.


**Strength And Weaknesses:**

### Strength
- The propose approach is novel on designing a hyper-prompt training  approach for neural retrieval out-of-domain generalization.
- The authors conduct comprehensive experiments and analysis to justify the effectiveness of every component of the proposed approach (hyper-prompt, contrastive prompt regularization). Results are shown on both multi-task and zero shot settings.

### Weakness
- This work claims to target the out-of-domain generalization for neural retriever but experiments only justify the out-of-dataset generalization. Most retrieval tasks are from the Wikipedia domain.


**Summary Of The Paper:**

The authors investigate the out-of-domain generalization problem in neural retrieval. They propose a hyper-prompt training mechanism and contrastive prompt regularization to this problem. The hyper-prompt training predicts a distribution over the predefined prompt parameters given the query, and get the task-adaptive prompt embeddings as prefix for the each layer of query encoder to augment the query representation. Contrastive prompt regularization solves the mode collapse problem of attention score distributions. Experimental results show that the proposed approach achieves slight improvement in the multi-task training setting but leads to significant improvement in zero shot setting and on out of domain tasks.

**Summary Of The Review:**

In summary, I feel that this is a good work in general, though there are some minor concerns on the definition of “out-of-domain generalization”.

---

> ### Author Response · Authors · 2022-11-15
> **Response to Reviewer tLxh**
>
>
> Thanks for your in-depth comments! We sincerely hope our replies could address your concerns.
>
> > **Question 1**: This work claims to target the out-of-domain generalization for neural retriever but experiments only justify the out-of-dataset generalization. Most retrieval tasks are from the Wikipedia domain.
> - Thanks for your insightful questions! In this paper, we focus on large-scale retrieval which aims to retrieve relevant documents from millions to billions of documents. In this area, Wikipedia is widely adopted as the retrieval knowledge base for many studies from different tasks (e.g., MS Marco from IR, NQ from QA, WoW from dialogue, etc.), since it contains high-quality documents, has a uniquely large scale and contains data across different domains. Therefore, we define the out-of-domain data as different tasks with different types of queries or the same task with queries from different domains, which includes 5 tasks and 7 datasets in KILT and 8 tasks and 10 datasets in BEIR.
> - Besides, if we use different retrieval knowledge bases, there is no need to build a uniform retrieval system since it causes the number of indexes to increase drastically. Moreover, the mixed knowledge bases (a larger index) may cause the degradation of performance.

---

### Official Review · Reviewer_X25B · 2022-10-26

**Confidence:** 4
**Correctness:** 3
**Technical Novelty And Significance:** 3
**Empirical Novelty And Significance:** 3
**Recommendation:** 8

**Clarity, Quality, Novelty And Reproducibility:**

The paper is clearly written and easy to read. Nice work.

I don't see any challenges in reproducing the results, but it's always nice to open source your codes and checkpoints.

**Strength And Weaknesses:**

The idea of computing query-dependent prefix to compute query vectors for retrieval is an appealing idea. The results shows that having learned prefixes does improve models' performance in many downstream tasks. The retrieval performance is also improved in zero-shot setting. Overall, I think this is an interesting paper. Thanks for the excellent work.

My main concern is the additional computation cost coming from the added prefixes. As discussed in the paper, the length of the prefix is 100. For many retrieval tasks, queries are short (sometimes 20~30 tokens). The added prefix tokens may increase the computation cost by ten times or more. Have you tried measuring the efficiency of your model and how it compares to the baselines?

If the improvement comes from having more parameters, why is it different from increasing the size of the model? Or, can you append 100 special tokens to queries and let models learn their embeddings.

A few more questions:
1. Does the number of task affect the size of the prefixes? Do you need more prefixes or longer prefixes if there are, say hundreds of tasks?
2. Are model's parameters in SPLADE jointly finetuned with parameters of the prefixes?

There are many paper come out recently in few-shot and zero-shot retrieval. You may want to add some of their numbers to your paper, e.g. Promptagator [1]. The numbers may not be directly comparable, but it's good to have.

[1] Promptagator: Few-shot Dense Retrieval From 8 Examples


**Summary Of The Paper:**

The paper proposed the idea of learning a dense retriever with a list of learned prefixes. The model attends to prefixes to compute query-dependent prefixes, which will be used to compute final query vectors for retrieval. In order to make the prefixes non-uniform, the paper additionally introduced a contrastive prompt regularization loss.

**Summary Of The Review:**

The proposed methods is well motivated and strongly justified by the experiments. The only drawback is that it's not very clear where does the improvement come from (see my question above). But overall, it is a nice paper.

---

> ### Author Response · Authors · 2022-11-15
> **Response to Reviewer X25B (2/2)**
>
>
> > **Question 3**: Does the number of task affect the size of the prefixes? Do you need more prefixes or longer prefixes if there are, say hundreds of tasks?
> - Thanks for your insightful questions!  Yes, the size of the prefixes is mainly related to the properties of tasks as indicated in [1,2].  We have also investigated in our paper that given a certain number of tasks, too few or too many prefixes result in suboptimal results. Intuitively, more tasks are involved in a unified framework, more parameters are needed to cluster them properly. Therefore, if there are hundreds of tasks, we may increase the number of prefixes to a proper length. However, to fully investigate this relation, much more experiments are required, we would like to leave this part in the future revision due to insufficient collection of data and limited computation resources.\
> &nbsp;
>   - [1] Li, Xiang Lisa and Percy Liang. “Prefix-Tuning: Optimizing Continuous Prompts for Generation.” ACL, 2021.
>
>   - [2] Zhao, Lulu et al. “Domain-Oriented Prefix-Tuning: Towards Efficient and Generalizable Fine-tuning for Zero-Shot Dialogue Summarization.” NAACL, 2022.
>
> > **Question 4**: Are model's parameters in SPLADE jointly finetuned with parameters of the prefixes
> - As mentioned in the Model Training section before Equation (8), we train the entire model for in-domain testing on KILT and we freeze the parameters of SPLADE for cross-domain zero-shot retrieval on BEIR.

---

> ### Author Response · Authors · 2022-11-15
> **Response to Reviewer X25B (1/2)**
>
> Thanks for your insightful comments! We will try our best to address your concerns and answer your questions.
>
> > **Question 1**: My main concern is the additional computation cost coming from the added prefixes. As discussed in the paper, the length of the prefix is 100. For many retrieval tasks, queries are short (sometimes 20~30 tokens). The added prefix tokens may increase the computation cost by ten times or more. Have you tried measuring the efficiency of your model and how it compares to the baselines?
> - We appreciate for pointing this out. As mentioned in our experiment setups, we have varied the length of the prefix and set it as 100, which is also adopted by many previous studies. Here, we further investigate how the length of prefixes influences the efficiency as suggested. To this end, we vary the size of the prefix in {5,10,20,50,100, 200} and record both the average time of encoding a query and retrieval performance on KILT. We compare our method with Prefix-tuning and the results are shown in the below table.  As we can see, our HypeR obtains better average retrieval scores than Prefix-Tuning for all different prefix lengths, and both Prefix-Tuning and HypeR achieve the best performance when the prefix length is 100. Notably, our HypeR shows significantly better average retrieval scores than the fine-tuning approach even when the prefix length is 10 while demonstrating acceptable encoding efficiency (0.25x growth). We have put the results as a figure in the appendix of our revision (Appendix C). Thank you again for your constructive suggestions.\
> &nbsp;
> |   Prefix Length | Method      |   Avg Scores |   Average Encoding Time |
> |----------------:|:------------|-------------:|------------------------:|
> |               5 | HypeR       |      59.8546 |               0.0292622 |
> |              10 | HypeR       |      60.1151 |               0.0322528 |
> |              20 | HypeR       |      60.2811 |               0.0410035 |
> |              50 | HypeR       |      61.2233 |               0.0678108 |
> |             100 | HypeR       |      61.38   |               0.078114  |
> |             200 | HypeR       |      61.0752 |               0.129016  |
> |               5 | Prefix      |      59.6546 |               0.0277832 |
> |              10 | Prefix      |      59.6709 |               0.0309543 |
> |              20 | Prefix      |      59.7611 |               0.03731   |
> |              50 | Prefix      |      60.03   |               0.0456711 |
> |             100 | Prefix      |      60.3733 |               0.0623451 |
> |             200 | Prefix      |      60.3512 |               0.119166  |
> |               0 | Fine-Tuning |      59.46   |               0.0258362 |
>
> > **Question 2**: If the improvement comes from having more parameters, why is it different from increasing the size of the model? Or, can you append 100 special tokens to queries and let models learn their embeddings.
> - Thanks for your suggestions! Actually, "append 100 special tokens to queries and let models learn their embeddings" is equal to the prefix-tuning baseline which has listed in Table 1/2/3. We can find that prefix-tuning performs worse than our Hyper on various settings and even longer prefixes may result in poor performance. The phenomenon implies that the improvement does not simply come from having more parameters. Instead, the improvement is mainly brought by our Query-conditional Prompt Synthesizer (QPS) which enables a neural retriever to better utilize the similarities between different tasks. Besides, our Contrastive Prompt Regularization (CPR) encourages different tasks to train and employ the collective skills among them, thus improving the overall performance. Hope the above clarification can address your concern.

---

> > ### Comment · Reviewer_X25B · 2022-11-15
> > **Followups**
> >
> > Thanks for your reply.
> >
> > From the results in the table, the average encoding time increases by 3 times, i.e. from 0.029 (5 tokens) to 0.078 (100 tokens). This is a significant change. With this computation budget, you can upgrade to a larger model, e.g. upgrade from BERT-base to BERT-large. What will be the improvement of performance of this upgrade?

---

> > > ### Author Response · Authors · 2022-11-16
> > > **Response to Reviewer X25B**
> > >
> > > We deeply appreciate your follow-up. First of all, we acknowledge that using a large model can improve performance in the fully supervised model. However, due to the limited time for rebuttal, we can not provide the detailed results of BERT-Large now. In this case, we would like to clarify that the main focus of our paper is to improve the generalization ability of a large-scale retrieval model, which is critical for building scalable IR systems. It has been widely proven that the performance of fine-tuning degrades significantly in the few-shot and zero-shot settings. Here we list the zero-shot performances of DPR (fine-tuning BERT) and our HypeR on the BEIR benchmark in the below table. We can find that our HypeR outperforms DPR by a large gap. Although scaling up the model size (e.g., BERT-Large) could bring improvement to the fully supervised setting, it can not avoid the intrinsic defects of traditional fine-tuning. In contrast, our method can achieve a balance between supervised performance and generalization performance.\
> > > &nbsp;
> > > | Model | TC   | NFC  | FiQA | Argu | Touche | DBP  | SciD | CFever | SciF | AVG  |
> > > |-------|------|------|------|------|--------|------|------|--------|------|------|
> > > | DPR   | 33.2 | 18.9 | 11.2 | 17.5 | 13.1   | 26.3 | 7.7  | 14.8   | 31.8 | 19.4 |
> > > | HypeR | 79.1 | 33.6 | 35.1 | 50.1 | 27.4   | 43.6 | 15.6 | 23.7   | 70.4 | 42.1 |
> > >
> > > Second,  the length of prefixes in our method controls the trade-off between efficiency and performance. With appropriate choices of the parameter, HypeR can be deployed to various scenarios with different computation budgets. For example, using a prefix consisting of 5 or 10 tokens can result in a considerable performance improvement. In this case, the encoding time only has increased by 13% or 25% while we have already owned superior transfer learning ability.  It may be worthwhile to achieve better performance in supervised settings as well as superior generalization ability with a reasonable increase in encoding cost.
> > >
> > > Nevertheless, we are very glad to investigate how to further improve the efficiency of our method in the future.
> > >
> > > Thanks again for your great question!

---

> > > > ### Comment · Reviewer_X25B · 2022-11-16
> > > > **Thanks for your clarification**
> > > >
> > > > Thanks. It is clear to me now.

---

### Decision · Program_Chairs · 2023-01-20

**Decision:**

Accept: poster

**Justification For Why Not Higher Score:**

Adding 100 extra soft-prompt tokens to the query reduces efficiency and may make the method less practical. The requirement of an additional contrastive loss somewhat complicates training.

**Justification For Why Not Lower Score:**

The method is well motivated and clearly explained. The approach shows convincing results when evaluated on the BEIR and KILT benchmarks, and outperforms relevant ablations such as prefix tuning. Overall, this is a good submission that should be accepted.


**Metareview: Summary, Strengths And Weaknesses:**

The submission introduces an architecture for improving dense retrievers in low resource settings. The approach learns a query-specific mixture of soft prompts which are used to improve a dense retrieval model. During training, an additional contrastive loss is used to prevent all tasks learning the same prompt.

The method tackles an important problem, and is well motivated and clearly explained. The approach shows convincing results when evaluated on the BEIR and KILT benchmarks, and outperforms relevant ablations such as prefix tuning.

Adding 100 extra soft-prompt tokens to the query reduces efficiency and may make the method less practical. The requirement of an additional contrastive loss somewhat complicates training.

Overall this is a good submission that should be accepted.

**Note From Pc:**

if the above contains the word "oral" or "spotlight" please see: "oral" presentation means -> notable-top-5% and "spotlight" means -> notable-top-25%. As stated in our emails, we are disassociating presentation type from AC recommendations

**Summary Of Ac-Reviewer Meeting:**

n/a